# Velvet Family Protein FpVelB Affects Virulence in Association with Secondary Metabolism in *Fusarium pseudograminearum*

**DOI:** 10.3390/cells13110950

**Published:** 2024-05-30

**Authors:** Yuxing Wu, Sen Han, Yajiao Wang, Qiusheng Li, Lingxiao Kong

**Affiliations:** Plant Protection Institute, Hebei Academy of Agricultural and Forestry Sciences, Integrated Pest Management Center of Hebei Province, Key Laboratory of IPM on Crops in Northern Region of North China, Ministry of Agriculture, Baoding 071000, China; wyx1209@haafs.org (Y.W.);

**Keywords:** *Triticum aestivum*, *Fusarium pseudograminearum*, velvet, conidiation, secondary metabolite

## Abstract

*Fusarium pseudograminearum* causes destructive crown disease in wheat. The velvet protein family is a crucial regulator in development, virulence, and secondary metabolism of fungi. We conducted a functional analysis of FpVelB using a gene replacement strategy. The deletion of *FpVelB* decreased radial growth and enhanced conidial production compared to that of wild type. Furthermore, FpVelB modulates the fungal responses to abiotic stress through diverse mechanisms. Significantly, virulence decreased after the deletion of *FpVelB* in both the stem base and head of wheat. Genome-wide gene expression profiling revealed that the regulation of genes by FpVelB is associated with several processes related to the aforementioned phenotype, including “immune”, “membrane”, and “antioxidant activity”, particularly with regard to secondary metabolites. Most importantly, we demonstrated that FpVelB regulates pathogen virulence by influencing deoxynivalenol production and modulating the expression of the *PKS11* gene. In conclusion, FpVelB is crucial for plant growth, asexual development, and abiotic stress response and is essential for full virulence via secondary metabolism in *F. pseudograminearum*.

## 1. Introduction

Fusarium crown rot (FCR), primarily caused by the soil-borne fungal pathogen *Fusarium pseudograminearum*, is a widespread and destructive disease that affects cereal crops, particularly wheat and barley [1]. This chronic soil-borne disease is notorious for its ability to inflict substantial yield losses and economic damage. Losses in yield due to this disease are as high as 10–35% under natural inoculum levels in Australia and the Pacific Northwest of the USA [2]. The prevalence of *F. pseudograminearum* has increased sharply in the Henan, Hebei, and Shandong provinces, which are the main wheat-producing areas in the Huang–Huai region of China [3,4,5]. Typical symptoms of FCR include crown, foot, and root rot. After infection, the coleopedia, bottom leaf sheath, and base stem of wheat show successive browning. Heavily infected plants produce white heads with no or few full seeds [6]. In addition to production loss, the pathogen can threaten human and animal health by contaminating food with the producing mycotoxins, such as deoxynivalenol (DON) and nivalenol (NIV) [7].

Plant pathogens can produce cell wall-degrading enzymes, secondary metabolites (SMs), and other pathogenic substances that participate in pathogenic diseases. *F. pseudograminearum* successfully infects and colonizes host tissues by producing certain SMs, such as the trichothecene toxin DON because the deletion mutants of the *TRI5* gene exhibit reduced virulence in wheat [8,9,10]. In addition, a gene cluster encoding a cytokinin-like metabolite has been identified in *F. pseudograminearum*. *Fusarium* cytokinins have been shown to activate cytokinin signaling in plants both in vitro and in vivo to help pathogens infect host plants [11]. In a recent study, the deletion of the nonribosomal peptide-coding gene FpNPS9 compromised DON production and virulence in *F. pseudograminearum*. Further analysis revealed at least 16 nonribosomal peptide synthetases and 14 polyketide synthases (PKSs) in the genome of *F. pseudograminearum*. These synthases play important roles in the secondary metabolic synthesis pathway [12]. Therefore, other SMs likely affect the pathogenicity of *F. pseudograminearum* [8,13]. In view of the important role of SMs in *F. pseudograminearum* infection, the study of these regulatory mechanisms of SMs may help the understanding of virulence mechanisms and provide an important reference for resistance breeding and disease control [7]. The regulatory mechanisms of SMs in fungi involve multiple processes such as signal transduction pathways and environmental factors. These regulatory factors can be divided into global regulators, pathway-specific modifications, and epigenetic modifications [14]. Among these, the light response-related velvet family proteins are important global regulators of SMs in fungi [15].

The velvet protein family comprises a class of fungal transcription factors sharing a common velvet domain. The four members—VeA, VelB, VelC, and VosA—have been well identified and characterized [16]. Among these, VeA was the most intensively studied protein that regulates development and SMs in a variety of fungi [15]. VelB, which can assemble into a heterotrimeric velvet complex with VeA, is involved in multiple regulatory processes in several filamentous fungi [17]. This complex plays an important role in conidiation and virulence by regulating the transcription factor VmCmr1, which is involved in melanin synthesis and in controlling the expression of pectinase genes in *Valsa mali* [18]. In addition, the lack of *VelB* hinders both conidiation and aflatoxin production in *Aspergillus flavus* [19]. VelB is crucial for the development, pathogenicity, and SM of *Penicillium expansum* [20]. VelB can also form a complex with VosA. This complex affects the growth, development, and SMs of *A. nidulans* in different ways and to different degrees [21,22,23]. The velvet complexes are present in the *Fusarium* genus. The VelB subunits within these complexes interact to control the biosynthesis of deoxynivalenol, fumonisin, and beauvericin in addition to governing aspects of the development, pathogenicity, and virulence of *F. graminearum*, *F. verticillioides*, and *F. oxysporum* [24,25,26]. In *F. pseudograminearum*, FpVeA, a key member of the velvet family, was identified via map-based cloning and regulates the virulence of *F. pseudograminearum* [27]. However, the function and regulatory mechanism of VelB remain poorly understood when SMs, key virulence factors, are regulated by VelB, as is the manner in which this regulation may occur.

In this study, we investigated the function of VelB orthologous protein, FpVelB, in the development, virulence, and expression of SM genes of *F. pseudograminearum*. The deletion of FpVelB led to notable differences in growth and conidiation. Furthermore, we observed reduced virulence in the *FpVelB* deletion mutants. FpVelB has also been identified as a key regulator of multiple metabolic pathway genes, particularly those associated with SMs. Notably, DON production depends on FpVelB. Additionally, FpVelB positively regulates another SM gene cluster associated with pathogenesis. These findings strongly suggest that FpVelB is crucial for growth, asexual development, and abiotic stress responses and is essential for full virulence via SMs in *F. pseudograminearum*.

## 2. Materials and Methods

### 2.1. Strains and Culture Conditions

Wild-type strains of *Fusarium pseudograminearum* 2035 were stored in the Laboratory of Fungi Diseases, Institute of Plant Protection, Hebei Academy of Agricultural and Forestry Sciences. All *F. pseudograminearum* strains were incubated on potato dextrose agar (PDA) medium (potato extract 200 g, dextrose 20 g, and agar 15 g per litre) at 25 °C.

The growth rates of *F. pseudograminearum* strains were calculated using the colony radius per day. Conidiation assays were performed using carboxymethyl cellulose sodium (CMC) medium [28]. The conidial concentration of the various strains was assessed using a hemocytometer following incubation in 100 mL CMC at 170 rpm and 25 °C for 4 d [29]. Complete medium (CM) supplemented with different inhibitors was used to assess the stress responses of the various strains. The vital medium consisted of yeast extract (6 g/L), peptone (2 g/L), casein hydrolysate (6 g/L), sucrose (10 g/L), NaNO_3_ (12 g/L), KCl (1 g/L), KH_2_PO_4_ (3 g/L), and agar (15 g/L). The inhibitors were supplemented with NaCl (0.7 M), 3 mM H_2_O_2_ (3 mM), Congo red (200 mg/L), or sodium dodecyl sulfate (SDS, 0.01%). TB3 medium, composed of yeast extract (3 g/L), casamino acids (3 g/L), sucrose (200 g/L), and agar (15 g/L), was used for the recovery and selection of resistant transformants for gene deletion or complementation procedures. Hygromycin B (250 μg/mL, Calbiochem, LaJolla, CA, USA) and geneticin (250 μg/mL, Sigma-Aldrich, St. Louis, MO, USA) were selected as screening markers for gene deletion or complementation, respectively.

### 2.2. Gene Deletion and Complementation

The *FpVelB* gene was initially examined by conducting homology searches for the genomic sequences of *F. pseudograminearum* (GenBank accession NC_031951.1) using the VelB protein of *A. nidulans* as a reference [17]. Alignment of FpVelB with homologous proteins was performed using ClustalW. Subsequently, a phylogenetic tree was constructed using the neighbor-joining method included in MEGA software version 7.02 [30]. Gene deletion constructs were created by substituting the full open reading frame (ORF) of target genes. The upstream and downstream flanking sequences of the target genes were amplified from the genomic DNA of wild-type strain 2035 using 1F/2R and 3F/4R primer pairs, respectively. Primer sets HYG/F and HYG/R were used to amplify fragments containing the partial hygromycin phosphotransferase gene. The three fragments were assembled using a double-joint overlapping PCR method [31]. The fusion replacement fragment was introduced into 2035 strain protoplasts and mediated using polyethylene glycol method. Protoplasts were prepared and transformed according to established procedures [32]. After screening with hygromycin B medium, hygromycin-resistant transformants underwent PCR screening with the primer pairs 5F/6R, H850F/H852R, 7F/H856R, and H855F/8R to validate the gene replacement events. Putative gene deletion mutants were validated through Southern blot analysis using an hph gene probe (designated as probe h). The procedure followed the manufacturer’s instructions (DIG-High Prime DNA Labelling and Detection Starter Kit II; Roche, Penzberg, Germany). The target gene fragments were amplified using primer pairs C-F/R from *F. pseudograminearum* genomic DNA. A gap repair method targeting gene fragments was used, and the XhoI-digested plasmid pFL2 was co-transformed into the yeast strain XK1-25 to create complementation strains [33,34]. Following obtained from the Trp+ yeast medium, the fusion constructs transformants were verified by sequencing and were subsequently introduced into the corresponding deletion mutant. Complementary transformants with genetic resistance were identified by PCR using the primer pairs 5F/6R. Appendix A contains a list of all primers utilized for deletion, complementation, and gene expression analyses.

### 2.3. Virulence Assays

The different strains were selected and inoculated on PDA plates for 3 days. Five agar plugs (5 mm each) taken from the colony’s edge were placed in a 250 mL Erlenmeyer flask containing 100 mL CMC. The Erlenmeyer flasks were stored at 170 r/min and 25 °C for 5 days. the conidial precipitate collected by centrifugation was suspended in sterile water at 10^5^ spores/mL. The Shixin 828 cultivar (a susceptible cultivar) with full grains was selected for infection testing. Seeds were disinfected using 1% sodium hypochlorite for 5 min. After washing with sterilized water three times, they were immersed in the configured conidial suspension for 5 min. A total of 20 immersed seeds were planted in 15 cm diameter pots containing sterilized soil. The pots were kept at a relative humidity of 60% ± 10% and a day/night temperature of 25/15 ± 5 °C. Disease severity was evaluated 21 days post inoculation (dpi) on a scale of 0 to 5 [35].

Floret injection was used for the wheat head infection test. The conidial suspension was cultured as described previously. At early anthesis, 20 µL conidial suspension was injected into a floret. Each treatment group was inoculated with 30 heads. Disease severity was evaluated at 20 dpi on a scale of 0 to 4 [36].

Both stem base and ear disease severities were represented as disease index (DI).

DI = [∑ (number of diseased plants in this scale × value of this scale)/(total number of plants investigated × highest value of the scale)] × 100.

### 2.4. RNA-Seq and qRT-PCR Analysis

Mycelial samples were collected from colonies cultured on PDA surfaces. Mycelial RNA was extracted using an RNA extraction kit (Qiagen, Hilden, Germany) according to the manufacturer’s instructions. Sangon Biotech Co., Ltd. (Shanghai, China), performed the library preparation and sequencing. The clean reads were aligned to *F. pseudograminearum* CS3096 reference genome using TopHat 2.0.8 software with default parameters [37]. Gene expression levels were determined using reads per kilobase per million (RPKM) reads of the mapped sequences. Differences in gene expression between the mutant and wild-type strains were assessed using HTSeq software (v0.9.1) [38]. Differentially expressed genes (DEGs) were determined using DESeq2 software (v1.12.4) with a false discovery rate (FDR)-adjusted *p*-value < 0.05 [39]. A log2 (fold change) exceeding 1.0, calculated based on the RPKM values of the same gene, signified a fold change between the mutant and wild-type strains. Gene ontology (GO) annotation was performed using GOseq package software (1.54.0) [40]. The analysis of Kyoto Encyclopedia of Genes and Genomes (KEGG) enrichment was achieved using clusterProfiler v3.8.1 software with a significance threshold of *p* < 0.05 [41].

Quantitative real-time PCR (qRT-PCR) was performed to assess the expression levels of genes related to SM [42]. The mutant and wild-type strains were inoculated on stem bases for 7 dpi, and stem base tissues were collected. Total RNA was extracted as described previously. First-strand cDNA was generated from total RNA using the Fermentas 1st cDNA Synthesis Kit (Hanover, MD, USA) according to the manufacturer’s guidelines. The expression levels were normalized using the β-tubulin (TUB) gene [43]. Results were computed using the 2^−ΔΔCT^ method [44]. Means and standard deviations were obtained from three biological replicates. Appendix A lists the primers used for gene expression analysis.

### 2.5. Determination of DON Production

Inoculation comprised the placement of three agar plugs of mycelium, each with a 6 mm diameter in 30 mL trichothecene biosynthesis induction medium in a 150 mL Erlenmeyer flask [45]. The Erlenmeyer flasks were agitated at 180 rpm and incubated at 28 °C for 14 d. After processing through a 0.22 μm aqueous filter, the ultra-performance liquid chromatography tandem mass spectrometry (UPLC-MS/MS) technique was employed to analyze the filtrate obtained from the fermentation broth [46].

### 2.6. Data Statistics

Fisher’s least significant difference test was used SPSS software (v26.0.0, IBM, Armonk, NY, USA).

## 3. Results

### 3.1. Construction of FpVelB Gene Deletion Mutant Strains

The *F. pseudograminearum* genome (accession number GCA_000303195.2) harbored a single copy of VelB, which was identified as *FpVelB* (accession number: XP_009262923.1).

The FpVelB protein, with 460 amino acids (aa), exhibited 46.72% homology with A. nidulans VelB. The positions aa 143–445 of this protein were predicted to constitute the velvet domain through a CD Search on NCBI. The gene encoding FpVelB was interrupted by five introns at positions 431–491, 548–603, 683–741, 1005–1062, and 1364–1427 bp. A phylogenetic tree was constructed using FpVelB and its orthologues from nine other filamentous fungi. The results showed that FpVelB is a conserved homologue of the velvet protein in filamentous fungi, closely related to *F. graminearum* (Figure 1A).

We generated null-mutants in which the entire ORF of *FpVelB* was deleted to explore the functions of the FpVelB genes in *F. pseudograminearum*. Gene replacement constructs with hygromycin resistance were transformed into 2035 wild-type strain protoplasts through polyethylene glycol-mediated protoplast transformation. Hygromycin-resistant transformants were confirmed by PCR using four primers and Southern blot analysis.

No product was detected when the ORF primer (FpVelB-5F/6R) was used for the two FpVelB deletion mutants (VDM-1 and VDM-2). The hygromycin gene was successfully detected, and the upstream and downstream recombinations were successfully amplified (Figure 1B). Additionally, single-locus homologous recombination events occurred in the two deletion mutants, as evidenced by the presence of a sole 5.3-kb fragment band hybridized with the hph probe (probe h) in the genomic DNA of *FpVelB* mutant strains (Figure 1C). Mutations in FpVelB were restored by reintroducing the wild-type allele at different genomic locations via genetic transformation, resulting in the generation of VDM-C strains (Figure 1D).

### 3.2. FpVelB Is Necessary for Normal Growth and Conidiation

We quantified the mycelial growth of different strains on PDA medium to assess the function of the *FpVelB* gene in the development of *F. pseudograminearum*. The growth rate of VDM-1 and VDM-2 were reduced. The mycelia of the two mutant strains became slender, and the colony color changed to yellow (Figure 2A,B). These findings suggest that FpVelB plays a crucial role in vegetative growth of *F. pseudograminearum*. We further determined the impact of the deletion of the *FpVelB* gene on conidiation. Conidial production increased in the induced culture medium after the deletion of the *FpVelB* gene (Figure 2C). This result indicated that *FpVelB* gene negatively regulated the conidiation of *F. pseudograminearum*. The reversal of both growth and conidiation phenomena was observed after reintroducing the *FpVelB* gene into the VDM-1.

### 3.3. FpVelB Affects Responses of F. pseudograminearum to Abiotic Stress

We examined the inhibition of growth rate in wild-type and mutant strains on CM supplemented with 0.5 M NaCl (osmotic pressure), 3 mM H_2_O_2_ (oxidative stress), 0.01% SDS (cell membrane-damaging agent), or 200 mg/L Congo red (a cell wall inhibitor; Figure 3A) to assess the involvement of FpVelB in abiotic stress response. These inhibitors led to varying degrees of vegetative growth inhibition in all the strains; however, significant differences were observed between the mutants and the wild type. The growth rate inhibition in VDM-1 and VDM-2 were more pronounced than in wild-type and complementary strains when exposed to 0.5 M NaCl, 3 mM H_2_O_2_, and 0.01% SDS media, respectively. After the deletion of *FpVelB*, the more sensitive to these three stressors, indicating that FpVelB was involved in resistance to osmotic stress, oxidative stress, and cell membrane integrity in *F. pseudograminearum*. The sensitivities of VDM-1 and VDM-2 were notably reduced when 200 mg/L Congo red was added to the medium (Figure 3B). This result shows that FpVelB negatively regulated the integrity of cell wall synthesis. In addition, the reintroduction of the *FpVelB* gene into the deletion mutant strain restored the observed phenomena to resemble those observed in the wild-type strain.

### 3.4. FpVelB Is Required for Full Virulence

To assess the potential role of FpVelB in disease development, virulence assays was conducted on the basal parts of the stem and flowering head of wheat. The lesion size was quantified, and the DI was calculated following artificial inoculation. At 20 and 21 days post inoculation, the wild-type plants exhibited typical symptoms of crown rot and head blight, respectively (Figure 4A). The *FpVelB* mutant strains led to a significantly reduced DI at the stem base and head compared to that caused by the wild-type strain (Figure 4B). To substantiate the role of FpVelB in *F. pseudograminearum* infection, the observation of the infection process was conducted within wheat coleoptile cells by dyeing with wheat germ agglutinin. The observations of the coleoptiles at the wheat seedling stage revealed that those from the wild-type inoculation were filled with mycelia, causing the extensive decomposition and breakage of most plant cells. In contrast, the coleoptiles of the mutant showed only a few mycelia, with the majority of cells remaining intact (Figure 4C). To validate these observations, a complementary study was conducted on the *FpVelB* gene. These results demonstrated that reintroducing *FpVelB* into the mutant strain rescued the previously observed phenotype. This clearly indicated the significant role of FpVelB in the virulence of *F. pseudograminearum*.

### 3.5. FpVelB Regulates Gene Expressions of SMs including DON

We conducted genome-wide transcriptome analyses (through RNA-seq) between the wild-type and *FpVelB* deletion mutant VDM-1 to elucidate the regulation of the metabolic pathway influenced by FpVelB (raw RNA-seq data can be found in the NCBI Sequence Read Archive under the accession number PRJNA1058845). A total of 1324 genes exhibited significant changes in expression levels in the deletion mutants of *FpVelB*. Upon conducting an expression analysis with a significance level of *p* < 0.05 and log2foldchange > 1 or <−1, a total of 834 genes were found to be downregulated, whereas the expression of 490 genes increased (Appendix A). A comprehensive GO database was used to elucidate the functions of DEGs. Approximately 60 terms are categorized into three main groups, i.e., “molecular function”, “cellular components”, and “biological process”. The DEGs enriched in cellular components were “membrane”, “membrane-enclosed lumen”, and “supramolecular fiber” categories, which may be associated with the involvement of FpVelB in the integrity of cell membrane and cell wall synthesis. In molecular function, the “antioxidant activity” function of DEGs was enriched (Appendix A). The GO analysis revealed that the most prominent functions of DEGs were enriched in immune-related metabolic pathways, including “response to host immune response”, “evasion or tolerance of host immune response”, and “active evasion of host immune response” (Appendix A). These enriched DEGs may be associated with the involvement of FpVelB in the pathogenicity of *F. pseudograminearum*. In the KEGG pathway analysis, the biosynthesis of SM DEGs was enriched. In total, 79 genes among 465 significant DEGs were annotated as associated with “secondary metabolites biosynthesis, transport, and catabolism” pathways (Figure 5A). The number of downregulated and upregulated genes was 47 and 32, respectively (Figure 5B). The transcriptional levels of the genes in different expression modes were confirmed using qRT-PCR to validate the accuracy of the transcriptomes. The results indicated that the relative expression of the five upregulated and five downregulated genes was consistent between the transcriptomes and qRT-PCR (Figure 5C).

Since DON is an important toxin in *F. pseudograminearum*, we assessed the transcriptional level of the *TRI5* gene and the production of DON in the mutants and 2035 strain under inducing medium conditions. The relative transcript level of *TRI5* was reduced by a factor of 50 in the VDM-1 and VDM-2 compared to that in the 2035 strain (Figure 5D). Moreover, in the VDM-1- and VDM-2-induced culture medium, the detections of DON were negligible, whereas the 2035 strain showed a DON level of 1143 at this point (Figure 5E). These findings imply that FpVelB is crucial for the regulation of DON production in *F. pseudograminearum*.

### 3.6. PKS11 Gene Regulated by FpVelB Is Involved in Virulence

In the comparison of gene expression between the *FpVelB* deletion mutant and the wild type, we found that the gene encoding the polyketide synthase PKS11 was downregulated. Twenty genes were identified within the SM gene cluster, containing PKS11. These genes were involved in biosynthesis, transport, and other functions (Appendix A). We examined the expression patterns of all genes within this gene cluster throughout the infection process. We observed that most genes within this gene cluster were downregulated during infection. Among these, the most significantly downregulated gene was *PKS11*, which exhibited nearly a 30-fold decrease in expression (Figure 6A). We constructed a deletion mutant of this key synthase gene (Figure 6B–D). The phenotypic outcomes indicated that knockout of the synthesis gene had no discernible effect on the growth of the pathogen but led to a reduction in pathogenicity (Figure 6E,F). This suggests that the regulation of FpVelB pathogenicity is linked to PKS11 or its gene clusters.

## 4. Discussion

Although velvet family proteins, such as VelB, have been characterized in numerous species, their functions in *Fusarium pseudograminearum* are unclear. Our findings revealed that *F. pseudograminearum* genome harbored a single copy of *VelB*. Furthermore, this gene is evolutionarily conserved across various fungal species, implying that velvet family proteins likely play crucial roles in fungi. We propose that FpVelB is crucial for growth. Among the closest relatives of *F. pseudograminearum* within the genus, the disruption of *FgVELB* also resulted in reduced hyphal growth. Colonies of the deletion mutants are bright yellow [24]. This result serves as robust validation of the present findings. The same effect on growth rate was observed in *F. fujikuroi* and *Botrytis cinerea*. *FfVEL2* and *BcVelB* deletion mutants display diminished aerial hyphal growth [47,48]. Nevertheless, the deletion of *VelB* in *Aspergillus flavus*, *V. mali*, and *Penicillium expansum* had no effect on the vegetative growth of these fungi. The accumulation of pigments continues to be affected [18,19,20]. These studies suggested that the functions of the velvet family of proteins differ among fungi.

Conidiation plays a crucial role in the infection and life cycle of plant pathogenic fungi in natural environments [49]. In the present study, FpVelB negatively regulated conidial production. This mode of action is similar to that of VelB in several fungi including *F. graminearum* [24], *V. mali* [18], and *B. cinerea* [48,50]. In contrast, VelB positively regulates conidiation in *A. nidulans* [51], *A. flavus* [19], and *P. expansum* [20]. Therefore, the regulation of conidiation by VelB is inconsistent among fungal species.

Response mechanisms to abiotic stress are critical for the survival of plant pathogens [52]. Regulation of different stresses is species-specific. In *F. verticillioides*, the transcription of the catalase-encoding gene *FvCAT2* was positively regulated by FvVelB. The resistance of *F. verticillioides* to oxidative stress is reduced when *FvVelB* is deleted [25]. Additionally, the lower basal accumulation of glycerol in *Curvularia lunata* leads to reduced resistance of the *ClvelB* mutant to stressors [53]. VmVelB positively regulates sensitivity to osmotic, oxidative, and cell wall inhibitor stresses in *V. mali* [18]. In the model fungus *A. nidulans*, VelB regulates cell wall synthesis due to the binding to the promoter region of the β-glucan synthase gene *fsA* [54]. The present study demonstrates that FpVelB positively regulates sensitivity to osmotic, oxidative, and cytomembrane inhibitor stresses. The regulatory effect on the cell wall synthesis inhibitor of FpVelB was negative. Because of the complex form of regulation by VelB, the potential mechanism needs to be further elucidated in *F. pseudograminearum*.

In pathogenic fungi, VelB can affect virulence through different mechanisms [48]. For example, this velvet protein plays a role in virulence because of its effect on oxalic acid production in *B. cinerea* [48,50]. The virulence of *M. oryzae* is also regulated by its participation in appressorial development [55]. Moreover, in *V. mali*, the regulation of pectinase levels by VmVelB leads to virulence [18]. The main mechanism by which VelB affects virulence is the regulation of SM. For example, the positive regulation of gibberellic acid biosynthesis and the negative regulation of bikaverin biosynthesis by FfVel2 are involved in the virulence of *F. fujikuroi* [47]. In *F. graminearum*, two crucial virulence factors, deoxynivalenol and trichothecenes, play important roles in virulence [24,56]. In the necrotrophic fungus, *P. expansum*, the involvement of VelB in pathogenicity may be related to the regulation of patulin, chaetoglobosin A, citrinin, and fumarylalanine [20]. In this study, we used genome-wide expression analysis to explore the reason for the reduction in virulence. The deletion of the FpVelB gene affected 10% of gene expression. There are twice as many downregulated genes as that of upregulated. In the KEGG pathway analysis, the biosynthesis of SM DEGs was enriched. In total, 79 genes among 465 significant DEGs were annotated as associated with “secondary metabolites biosynthesis, transport and catabolism” pathways. This regulatory result is similar to that seen in *P. expansum*, where the *PeΔvelB* strain showed 1180 DEGs, of which 533 were up-expressed and 647 down-expressed. The expression of several backbone genes, including non-ribosomal peptide synthetases (NRPSs), polyketide synthases (PKSs), and dimethylallyl tryptophan synthases (DMATSs) was different in the Pe*ΔvelB* strain [20]. More importantly, we demonstrated that FpVelB not only influences virulence by regulating DON production but also participates in virulence by regulating other secondary metabolic synthesis pathways such as PKS11. Fungal PKSs are important in various cellular processes including cellular growth and development, environmental adaptation, and virulence in several pathogenic fungi. PKS11, a member of the PKS family, has been shown to be involved in responses to oxidation, UV irradiation, high temperature, asexual development, and cell wall integrity in *Beauveria bassiana*. In *P. marneffei*, the fungal virulence was reduced when *pks11* was knocked down [57]. In this study, the *PKS11* gene was removed from the FpVelB regulatory gene, and the deletion of *PKS11* gene led to the weakening of pathogen virulence, indicating that FpVelB involved in the virulence by regulating PKS11. The result provides a basis for understanding the pathogenesis of *F. pseudograminearum*.

## 5. Conclusions

In conclusion, the results of this study show that the velvet family protein FpVelB regulates vegetative growth, conidiation, and abiotic stress responses in *F. pseudograminearum*. Moreover, the regulation of virulence by FpVelB was associated with SMs such as DON and the expression of polyketide synthase gene PKS11.

## Figures and Tables

**Figure 1 cells-13-00950-f001:**
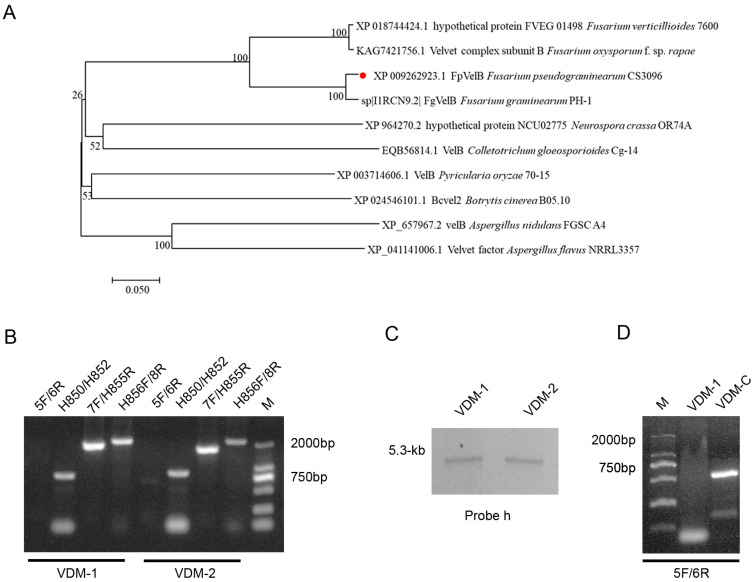
Phylogenetic analysis and mutant construction of FpVelB. (**A**) MEGA6 was used to compare the homologous genes of FpVelB (Marked with a red dot) in *Fusarium pseudograminis* with those in other fungi using neighbor-joining analysis with 1000 bootstrap replicates. The numbers on the branches indicate the percentage of replicates. The bar indicates a sequence divergence of 20%. (**B**) The target gene, *hph*, was denoted by the product of four primer pairs (5F/6R, H850F/H852R, 7F/H855F, and H856F/8R), representing the recombination of upstream and downstream regions, respectively. (**C**) The Southern blots show that genomic DNA of Δ*FpVelB*, digested with HindIII, exhibited bands that hybridized with the probe h. (**D**) The target fragment was verified by primer 5F/6R to prove the successful construction of the complementary strain VDM-C.

**Figure 2 cells-13-00950-f002:**
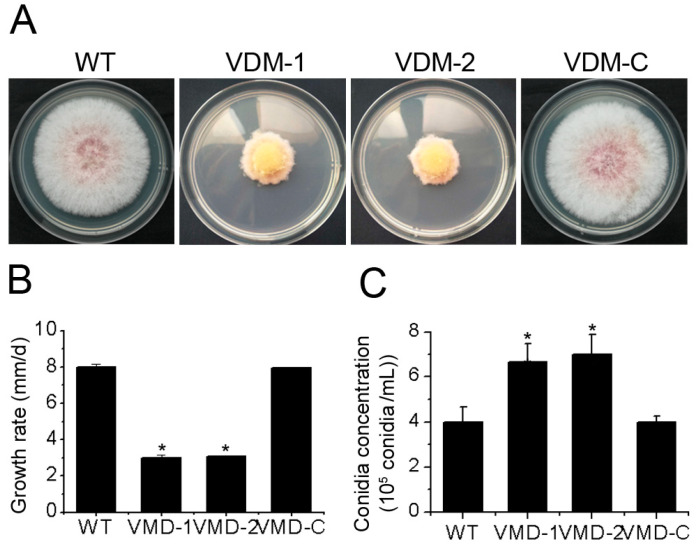
Effects of *FpVelB* on growth and conidia of *F pseudograminearum*. (**A**) Colony morphology of wild type (WT), mutants (VDM-1 and VDM-2) and complemented (VDM-C) strains were observed after three days culture. (**B**) Growth rates were calculated by measuring the variance in radial growth between two and three days post-inoculation (dpi, growth radius per day). (**C**) The logarithm of the conidia number per milliliter was assessed in the induced medium four dpi. Mean and standard are given (*n* = 3). Asterisks denote a significant difference compared to the wild type (*p* < 0.05).

**Figure 3 cells-13-00950-f003:**
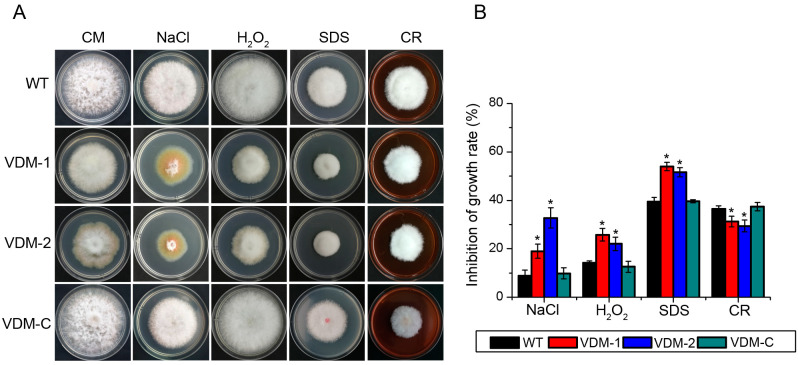
Impact of *FpVelB* gene inactivation on the stress responses of *F. pseudograminearum*. (**A**) The wild type (WT), mutants (VDM-1 and VDM-2), and complemented (VDM-C) strains were grown on CM. Three days post-inoculation (dpi), the colony morphology on various stress media was recorded. (**B**) The inhibition rates under different stress conditions were compared to that in CM without inhibitors. The bars represent the standard deviations of the mean calculated three replicates. An asterisk indicates a statistically significant difference compared to the wild type (*p* < 0.05).

**Figure 4 cells-13-00950-f004:**
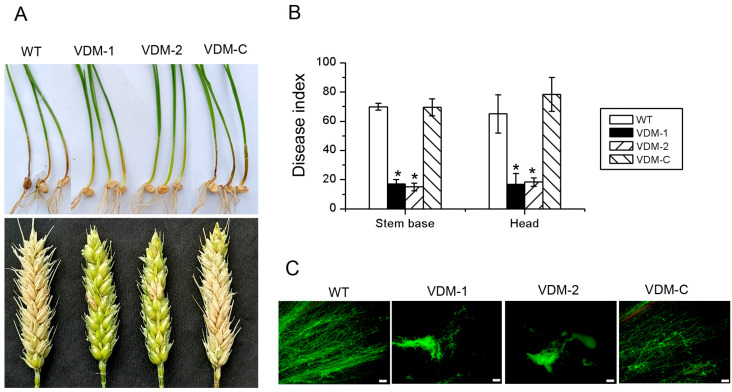
Phenotypes of stem bases and heads inoculated with *FpVelB* deleted mutants. (**A**) FpVelB mutant and its derivative strains were inoculated into the stem base and flowering heads of wheat to analyze its pathogenicity. (**B**) The disease index of stem base and ear of wheat in three test repeats were measured at 21 dpi and 20 dpi, respectively. An asterisk indicates a significant difference compared to the wild type (*p* < 0.05). (**C**) The infected coleoptiles were dyed using wheat germ agglutinin (WGA). Micrographs of different strains on the base of the wheat stem were captured at 21 dpi. Scale bar = 100 µm.

**Figure 5 cells-13-00950-f005:**
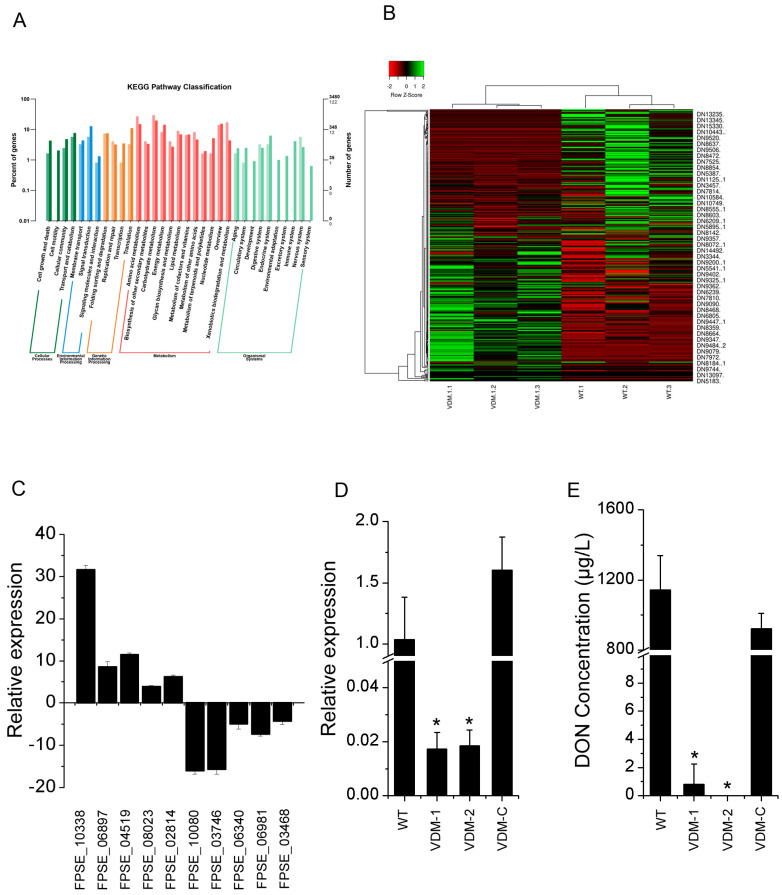
Differences in gene expression and DON production between FpVelB deletion mutant and wild type. (**A**) The DEGs were categorized by Kyoto Encyclopedia of Genes and Genomes (KEGG) pathway annotation. The DEGs were in log2 (fold change, FC) greater than 1.0 with a threshold at *p*-value and corrected *p*-value < 0.05. (**B**) Heatmap illustrating differentially regulated genes encoding secondary metabolites (SMs). (**C**) qRT-PCR was performed for ten differentially regulated genes, comprising five upregulated genes and five downregulated genes, between *FpVelB* mutants and the wild-type. The expression level of the TUB gene was used to normalize samples. Transcript levels of the wild type were arbitrarily set to 1. (**D**) Relative transcript abundances of the *TRI5* gene in mycelium under inducing conditions were compared between the wild type and *FpVelB* mutants at seven days post-inoculation (dpi). The expression level of the TUB gene was used to normalize different samples, with the transcript levels of the wild type arbitrarily set to 1. (**E**) The concentration of deoxynivalenol (DON) in the culture solution under inducing conditions was measured. The means and standards are given (*n* = 3). The data from these replicates were then analyzed using a protected Fisher’s least significant difference (LSD) test * *p* < 0.05.

**Figure 6 cells-13-00950-f006:**
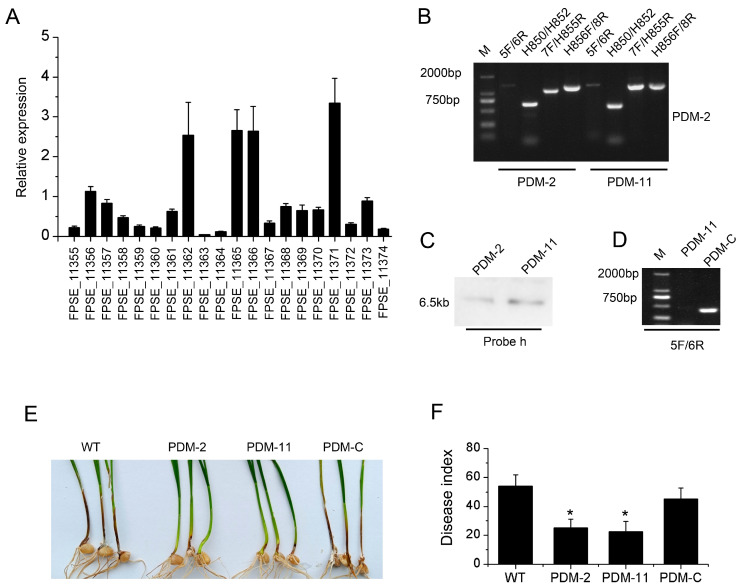
Creating mutants with deleted genes and testing the virulence of the *PKS11* gene in *F. pseudograminearum*. (**A**) Conducting qRT-PCR to analyze the expression of genes within the SM gene cluster, including *PKS11*, while infecting the wheat stem base in real-time. Samples were collected three days post-inoculation. Transcript levels of the conidia inoculator were arbitrarily set to 1. The means and standards are given (n = 3). (**B**) The *PKS11* gene, hph, was designated as the product of four primers (5F/6R, H850F/H852R, 7F/H855F, and H856F/8R), representing the recombination of upstream and downstream regions, respectively. (**C**) The Southern blots revealed that the genomic DNA of Δ*PKS11*, digested with HindIII, exhibited bands that hybridized with the probe h. (**D**) The successful complementation was confirmed by detecting the 5F/6R product in PDM-C. (**E**) FpVelB mutant and its derivative strains were inoculated into stem base of wheat, which was replicated three times. Images were captured at 21 days. (**F**) The disease index of the wheat stem base was assessed 21 days post-inoculation (dpi). An asterisk indicates that the mutant strain is significantly different from the wild type (*p* < 0.05).

## Data Availability

The original contributions presented in the study are included in the article/Appendix A. Further inquiries can be directed to the corresponding author.

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
