# Peer review of "Velvet Family Protein FpVelB Affects Virulence in Association with Secondary Metabolism in Fusarium pseudograminearum"

_cells, 2024, doi:10.3390/cells13110950_

Round 1

Reviewer 1 Report

Comments and Suggestions for Authors

This study demonstrates the prominent role of velvet family protein FpVelB in regulation of vegetative growth, conidiation, and abiotic stress responses in F. pseudograminearum. The work is meticulously performed and methodology is proper and with significant results.

1.      Please provide the accession details of the Wild-type strains of Fusarium pseudograminearum 2035. Were these identified using one of two conserved genes.

2.      A representative picture for construct will also be helpful in methodology.

3.      Please correct the grammatical errors across the manuscript. Please modify lines 10-12 for better clarity about the work and likewise do corrections.

4.      Line 27-29 require citation. And many factual sentences lack citations. Latest citations are less.

5.      Please delete or modify nonscientific words “insidious, and such similar words across the manuscript.

Comments on the Quality of English Language

minor editing required

Reviewer 2 Report

Comments and Suggestions for Authors

The concept of the manuscript is articulated elegantly, and the research encompassed within is meticulously designed. The experiments are executed with precision, and the exposition is clear. Upon my evaluation of the manuscript, I identified no issues. Nonetheless, I would like to suggest a singular improvement: Integrating hierarchical clustering of the samples in the heatmap depicted in Figure 5B could substantially enhance the analytical depth. Such clustering would illuminate the similarities among the expression profiles of different samples, thereby offering more profound insights into the dataset.

Reviewer 3 Report

Comments and Suggestions for Authors

This paper is well organized and easy to follow. In general, it’s a nice study and suitable to publish in Cells before minor revision.

1)      Line 61-62, “The velvet protein family comprises a class of fungi-specific proteins that respond to light signals. The family comprises four members: VeA, VelB, VelC, and VosA.”

These two sentences may be revised. First, the definition of velvet protein family is not fully correct. Second, this family comprises many more members. I would like to recommend a paper for you.  Phylogenomics analysis of velvet regulators in the fungal kingdom (https://journals.asm.org/doi/10.1128/spectrum.03717-23)

2) “3.6 PKS11 Regulated by FpVelB Is Associated With Virulence in F. pseudograminearum” Based on current data in the manuscript, I think it is hard to make such a conclusion. Do you know which product PKS11 responsible for? It has many reports that some SMs in Fusarium related with virulence.
